# Trace element catalyses mineral replacement reactions and facilitates ore formation

Yanlu Xing [1,2✉], Joël Brugger [1✉], Barbara Etschmann[1], Andrew G. Tomkins [1], Andrew J. Frierdich [1] & Xiya Fang[3]

Reaction-induced porosity is a key factor enabling protracted fluid-rock interactions in the Earth's crust, promoting large-scale mineralogical changes during diagenesis, metamorphism, and ore formation. Here, we show experimentally that the presence of trace amounts of dissolved cerium increases the porosity of hematite ($Fe_2O_3$) formed via fluid-induced, redox-independent replacement of magnetite ($Fe_3O_4$), thereby increasing the efficiency of coupled magnetite replacement, fluid flow, and element mass transfer. Cerium acts as a catalyst affecting the nucleation and growth of hematite by modifying the $Fe^{2+}(aq)/Fe^{3+}(aq)$ ratio at the reaction interface. Our results demonstrate that trace elements can enhance fluid-mediated mineral replacement reactions, ultimately controlling the kinetics, texture, and composition of fluid-mineral systems. Applied to some of the world's most valuable orebodies, these results provide new insights into how early formation of extensive magnetite alteration may have preconditioned these ore systems for later enhanced metal accumulation, contributing to their sizes and metal endowment.

[1] School of Earth, Atmosphere and Environment, Monash University, Melbourne, VIC, Australia. [2] School of Earth and Environmental Sciences, University of Minnesota, Minneapolis, MN, USA. [3] Monash Centre of Electron Microscopy, Monash University, Melbourne, VIC, Australia. ✉email: yanluxxing@gmail.com; joel.brugger@monash.edu

Ore deposit formation, the associated fluid-induced alteration, and large-scale metasomatic processes such as dolomitization and serpentinization, require efficient advection of fluids, volatiles, and/or metals over length scales of metres to hundreds of kilometres. Since intrinsic rock permeability is commonly low within the lithosphere[1], understanding the formation and destruction of interconnected fluid pathways is required to explain large-scale metasomatism. Deformation processes such as faulting and hydraulic fracturing provide interconnected fluid pathways at the macro-scale[2]; away from these, microfractures and grain boundaries were assumed to be the main fluid pathways until it was realised that porosity is intrinsically generated during many fluid-mediated mineral replacement reactions[3]. These reactions involve dissolution of the parent mineral coupled in space and time with precipitation of the daughter mineral and co-generation of porosity (microfractures, nano- to macro-pores) that allows ongoing mass transfer between the reaction front and the connected fluid network. Reaction kinetics, rather than equilibrium, is the main control on the evolution of these systems[4,5].

Magnetite and hematite are the most widespread iron oxide minerals in the Earth's crust and are present in most rocks. Economic iron deposits mostly consist of large accumulations of these minerals. In some important deposit classes, including iron oxide copper gold (IOCG) and rare earth element (REE) deposits, it is commonly observed that early magnetite is replaced by hematite in the main ore-forming stage[6–9], and voluminous iron oxide-rich ores are associated with enrichments in light REE (LREE), in particular, Ce (refs. [7,10,11]). In hydrothermal fluids, the main Ce oxidation state is Ce(III)[10]. An association between Ce(III) oxidation and Fe-oxide minerals is well established at low temperature, for example, preferential Ce removal from seawater results from adsorption of dissolved Ce(III) onto Fe and Mn oxides followed by oxidation of the surface complex[12]. This suggests that Ce(III/IV) may be involved in magnetite-to-hematite transformation reactions. To test this hypothesis, we performed hydrothermal experiments on the magnetite-to-hematite transformation in Ce-bearing and Ce-free solutions. For the Ce-free solutions, the non-redox active REEs La and Nd were added for comparison. For both sets of experiments, oxidant-free and $O_2$-rich solutions were used to test the influence of bulk fluid redox[4,13] (details in Methods).

We report new experimental results that demonstrate the hitherto unrecognised effect of trace elements in controlling the nature of porosity during fluid-induced mineral reactions. Specifically, we show how trace amounts of Ce(III) in solution result in generation of macro-porosity instead of nano-pores during the hydrothermal replacement of magnetite by hematite, thereby catalysing the replacement process. Using examples of giant ore deposits that feature widespread hematitisation of pre-existing magnetite (Olympic Dam, the archetypal IOCG deposit, and Bayan Obo, the world's largest REE deposit), we compare experimental and natural ore textures, investigate how the catalytic effect of Ce on hematitisation may have contributed to ore formation, and discuss the general implications for ore formation models.

## Results

### Cerium influences the replacement of magnetite by hematite.

Experiments conducted in $O_2$-rich solutions produced significantly different results compared to solutions devoid of an oxidant. For samples from $O_2$-rich runs, little reaction was observed regardless of the presence of Ce. The reacted magnetite grains in these experiments have smooth surfaces with pores usually isolated and small (<1 μm, Supplementary Fig. 1). Synchrotron X-ray fluorescence mapping (XFM) combined with μ-XANES imaging shows that Ce within the newly formed hematite

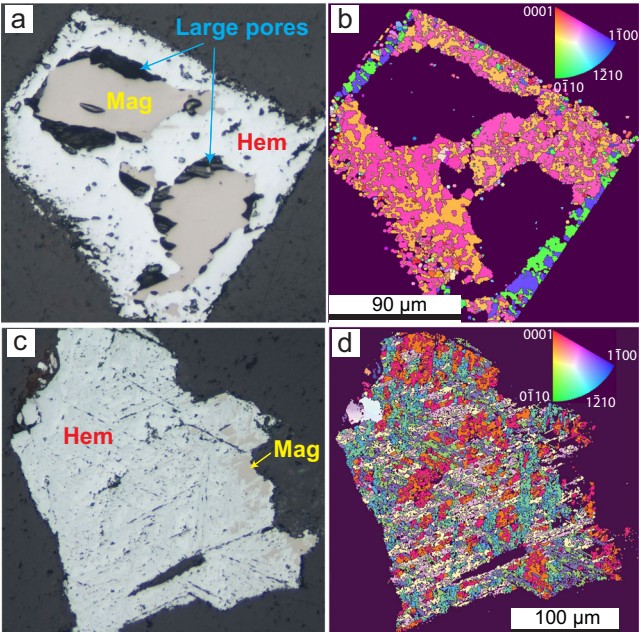

**Fig. 1 Effect of cerium on the replacement textures. a, c** Optical reflected light micrographs and **b, d** EBSD orientation maps. The reactions were conducted in oxidant-free Ce-bearing solutions (**a, b**) and in oxidant-free Nd-bearing solutions (**c, d**). **a** Hematite replaces magnetite along the rim; gaps occur at the hematite–magnetite interface; large (5–30 μm) pores occur within hematite. **b** EBSD inversed pole figures map showing hematite crystal orientation and boundaries in **a**. **c** Hematite replaces magnetite from one side of the parent magnetite grain. Hematite forms lamellae that are in close contact with magnetite, with no visible pores at the interface. Small (<1–10 μm) pores are present within the hematite. **d** EBSD inversed pole figures map showing hematite crystal orientation and grain boundaries in **c**.

rim is present mainly as Ce(IV), whereas only minor amounts of Ce(III) are present (Supplementary Fig. 2). Hence, under highly oxidizing conditions, Ce(III) is nearly fully oxidized to Ce(IV), at least partly within the solution itself via reaction with $O_2$(aq)/$H_2O_2$(aq). The precipitation of Ce in the form of insoluble $CeO_2$(s) prevented active involvement of Ce(III)/Ce(IV) in the hematitisation reaction. The smooth grain surfaces observed in these experiments (Supplementary Fig. 1) and the slow reaction rate[4] (Supplementary Table 1) indicate suppressed fluid–mineral interactions, which most likely reflects the reduction of porosity as a result of the volume increase associated with the oxidation of magnetite (see reaction (1) below).

Extensive magnetite replacement by hematite was observed when the reaction was conducted in oxidant-free solutions (Fig. 1a, b; Supplementary Table 1). Importantly, the products formed in Ce-doped solutions have different textures from Ce-free solutions (Fig. 1). In Ce-doped solutions, magnetite was replaced by hematite rims, with 5–10 μm wide gaps usually developed at the hematite–magnetite phase boundary (Fig. 1a). Secondary electron (SE) images show that the magnetite grains have rough surfaces, through formation of hematite, with significant variance in crystal sizes (1 to >5 μm; Fig. 2a). By comparison, in experiments using La- or Nd-doped solutions, hematite usually forms thin lamellae in close contact with magnetite (Fig. 1c); hematite crystals are closely stacked on the magnetite surfaces and have a homogenous morphology with a narrow size distribution (~5 μm; Fig. 2b).

Electron backscattered diffraction (EBSD) analyses were performed to determine crystal orientation and crystal dimension

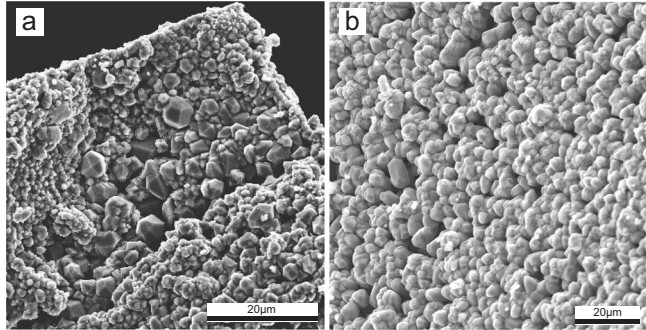

**Fig. 2 Secondary electron image of the sample surface.** Magnetite grains replaced by hematite, for reactions in oxidant-free conditions and in the presence of either **a** Ce(III) or **b** Nd(III) in solution. In **a**, recrystallized hematite shows significant variance in crystal sizes, whereas in **b**, crystal sizes are relatively invariant.

statistics (e.g. crystal morphology and size distribution) for hematite. In both scenarios (Fig. 1b, d), the transformation obeys the topotaxial and epitaxial relationships that are well described for hematite replacing magnetite, with crystal orientation $\{111\}_{mt}$ parallel to $\{0001\}_{hm}$ (ref. [14]); this also results in three orientations of hematite via replacement of a single crystal magnetite (Supplementary Fig. 3). However, in Ce-doped solutions, hematite grains generally grow larger than those in La- or Nd-doped solutions. EBSD inversed pole figures maps and pole figures further show that in Ce-doped experiments, neighbouring grains tend to have similar orientations (Fig. 1b; Supplementary Fig. 3), resulting in the formation of large domains with near-consistent orientations. In contrast, neighbouring grains show less correlation regarding their relative orientations in the Ce-free experiments (Fig. 1d; Supplementary Fig. 3). Overall, the observed hematite porosity (size and distribution) and crystal habit indicate that Ce substantially influences the dissolution-reprecipitation process[15].

**Reaction mechanism.** The replacement of magnetite by hematite can happen via oxidative or redox-independent pathways, characterised respectively by a small volume increase and by a large volume decrease[4,13]:

$$2Fe_3O_4(mt) + 0.5O_2(aq) = 3Fe_2O_3(hm);$$
$$\Delta V = 1.66\% \quad \text{(oxidative)} \tag{1}$$

$$Fe_3O_4(mt) + 2H^+ = Fe_2O_3(hm) + Fe^{2+}(aq) + H_2O;$$
$$\Delta V = -33\% \quad \text{(redox} - \text{independent)}. \tag{2}$$

In oxidative experiments, the product texture indicates that the overall transformation reaction can be described by simple oxidation (reaction (1)). The volume increase likely explains the low porosity and fracture annealing (Supplementary Fig. 1), and thus the slow reaction rate[4] (Supplementary Table 1).

For experiments conducted in oxidant-free solutions, the product textures indicate a typical interface-coupled dissolution-reprecipitation (ICDR) reaction mechanism[4]. In this process, the dissolution of magnetite can be described by

$$Fe_3O_4(mt) + 8H^+ = Fe^{2+}(aq) + 2Fe^{3+}(aq) + 4H_2O, \tag{3}$$

while hematite reprecipitation can proceed via the following reactions:

$$2Fe^{3+}(aq) + 3H_2O = Fe_2O_3(hm) + 6H^+, \tag{4}$$

$$2Fe^{2+}(aq) + 3H_2O = Fe_2O_3(hm) + 4H^+ + H_2(g). \tag{5}$$

When Ce(III) is present in solution, the following reaction may occur:

$$Fe^{3+}(aq) + Ce^{3+}(aq) + 2H_2O = CeO_2(s) + Fe^{2+}(aq) + 4H^+, \tag{6}$$

for which the stability constant (log $K$) is 0.93 (based on the SUPCRT database[16], updated with the properties of $CeO_2$(s) of Konings et al.[17]) at 200 °C. Assuming that the fluid contains 200 ppm Ce(III) at pH ~4, the equilibrium $Fe^{2+}$(aq)/$Fe^{3+}$(aq) ratio at the reaction front is $10^{13}$ at 200 °C. Since this value is much larger than the expected $Fe^{2+}$(aq)/$Fe^{3+}$(aq) ratio of 0.5 resulting from magnetite dissolution at the reaction front (reaction (3)), the presence of Ce(III) can dramatically increase the $Fe^{2+}$(aq)/$Fe^{3+}$(aq) ratio (>>0.5) in the local solution via reaction (6), as well as the solution acidity, promoting local magnetite dissolution, and increasing dissolution kinetics. Furthermore, as Fe(II) complexes are more stable than Fe(III) complexes in hydrothermal fluids[18], the increased $Fe^{2+}$(aq)/$Fe^{3+}$(aq) ratio would favour Fe mobilisation in local fluids. Conversely, reactions (4) and (5) show that hematite precipitation is hindered at low pH and/or under reducing conditions (i.e. high $Fe^{2+}$/$Fe^{3+}$ in solution)[19]. These conditions are favoured by local Ce(III) oxidation following reaction (6).

Reaction (6) results in the formation of $CeO_2$(s) [cerianite-(Ce)], which is supported by the XANES imaging results, confirming the presence of Ce(IV) in the hematite product (Supplementary Fig. 2). Cerium(IV) is poorly soluble in hydrothermal fluids and will precipitate as $CeO_2$(s). XANES results show that only small amounts of Ce(IV) are preserved in the hematite product, whereas Ce(III) is predominant (Supplementary Figs. 2 and 4); hence, we suggest that $CeO_2$(s) is recycled back to soluble Ce(III) via the following reaction:

$$CeO_2(s) + 3H^+ + 0.5H_2(g) = 2H_2O + Ce^{3+}(aq). \tag{7}$$

This recycling process explains how a trace element such as Ce can catalyse a major reaction, and tallies with the fact that cerianite-(Ce) is not reported as a mineral in IOCG ores. Reaction (7) shows that decreased pH, induced by hematite precipitation (reactions (4) and (5)) and locally reducing conditions at the hematite–magnetite interface (e.g. $fH_2$(g) = $10^{-3}$ at 200 °C, $P_{sat}$ for the hematite–magnetite buffer), will promote $CeO_2$(s) decomposition (Supplementary Fig. 5), and therefore help to maintain Ce(III) concentrations in solution.

In summary, the increase in Fe solubility at the reaction front in the presence of Ce(III) decreases the hematite nucleation rate. The interconversion between Ce(III) and Ce(IV) corresponds to subtle changes in pH and redox at the magnetite–hematite interface (Supplementary Fig. 5), which enable Ce to be actively involved in the magnetite–hematite transformation reaction but also keeps Ce concentration relatively stable, indicating its catalytic role (Fig. 3).

**Nature of reaction-induced porosity.** Reaction-induced porosity is important for mineral replacement reactions to proceed via ICDR[20], as it allows chemical exchange between the reactant and the bulk fluid[3]. This porosity is commonly transient, and not preserved as the system undergoes further fluid–rock interaction or annealing[5]. The amount of porosity is controlled by relative mineral solubilities, which in turn are controlled by fluid parameters such as pH, Eh, and solution composition[5]. It has recently been pointed out that ICDR is important in ore deposit formation[21], so identifying the key factors affecting porosity generation during ICDR reactions is fundamental to our understanding of ore genesis, and thus mineral exploration and extraction.

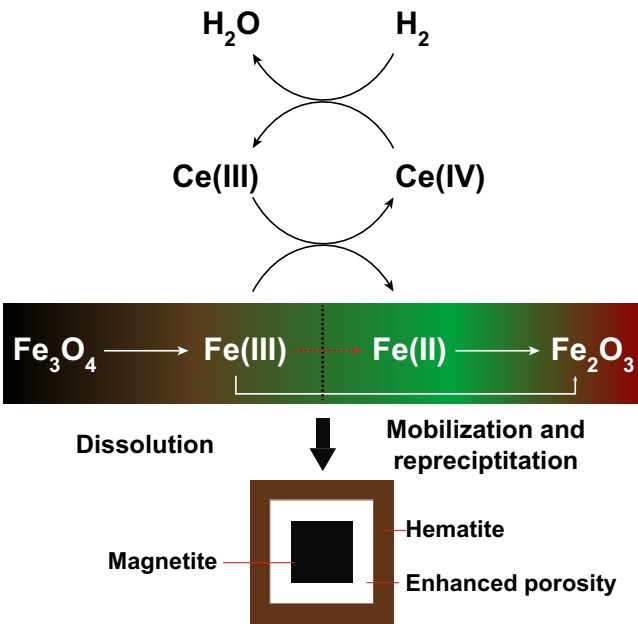

**Fig. 3 Fe(III)-Fe(II) and Ce(III)–Ce(IV) reaction cycles.** The presence of Ce(III) affects the Fe(III)–Fe(II) interconversion and contributes to enhanced porosity at magnetite–hematite interface.

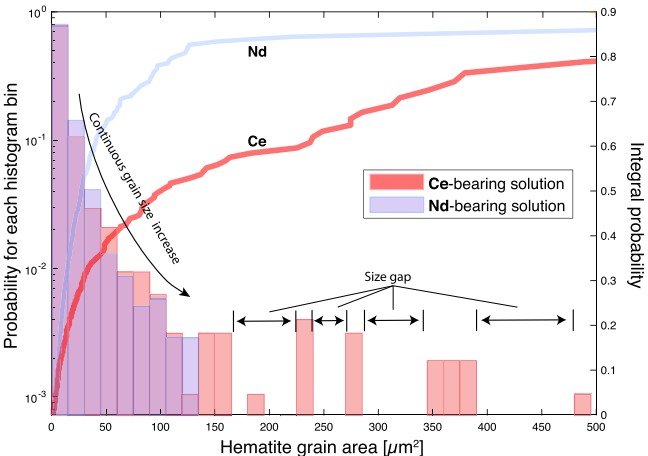

**Fig. 4 Grain size distribution of newly formed hematite.** The figure shows the crystal size of hematite formed in Ce-bearing and in Nd-bearing solutions from non-oxidative runs. The lines show the integral probability of hematite grain size. The probability of finding larger grain sizes decreases up to 150 µm$^2$ in both Ce and Nd-bearing solutions; however, a significant amount of hematite in the Ce-bearing solutions occurs as crystals larger than 150 µm$^2$, with large gaps in observed crystal sizes.

Rather than fine pores distributed throughout hematite (Fig. 1c), our oxidant-free Ce-bearing experiments produced coarse porosity located predominantly at the magnetite–hematite phase boundary (Fig. 1a). EBSD data confirm the development of large pores between large hematite crystals when Ce is present (Fig. 4). These are key changes that enhance fluid access (advective rather than diffusive transport) to the reaction front, thereby facilitating fluid–mineral interaction. The coarser and more connected porosity makes fluid penetration increasingly efficient compared to homogeneously distributed fine pores (Fig. 1c), as illustrated by the greater reaction extent in the presence of Ce compared to Ce-free experiments (runs MH39/MH44 vs. MH38/40/45; Supplementary Table 1). In the presence of Ce(III) in solution, hematite precipitation is the rate-limiting step in the magnetite replacement reaction (relatively slow nucleation), which allows formation of larger hematite crystals with concomitantly larger spaces between crystals, facilitating porosity generation in the iron oxide ores. In contrast, in the absence of Ce, hematite nucleation is fast and magnetite dissolution is the overall rate-liming step; this produces finer hematite crystals and more homogeneously distributed, smaller pores (Figs. 2, 4; Supplementary Fig. 1). The closer proximity of magnetite and hematite in this case (Fig. 1c) makes fluid access to the mineral reaction front less efficient.

## Discussion

Hematitisation of pre-ore magnetite is a key feature in some of the world's richest ore deposits. Examples include Olympic Dam, the world's largest uranium resource, the fifth largest copper, and the third largest gold resource[11], and Bayan Obo, the largest REE deposit in the world[22]. In both cases, magnetite is commonly replaced by porous hematite in the main stage ores[10,22]. It is further noted that in deposits throughout the giant Olympic Dam IOCG province in South Australia, the ores are hosted in hematite-rich breccia, and ore grade, brecciation intensity, and hematite:magnetite ratio are positively correlated[23,24]. Mineralized samples show increased porosity in hematite compared to magnetite[25], suggesting that the magnetite-to-hematite transformation process introduced porosity.

Focusing on the Olympic Dam IOCG province, a two-stage model was previously proposed to explain the association of the Cu–Au mineralisation with hematite alteration based on observations of subeconomic prospects in the region[26]. Early high temperature (>400 °C) hydrothermal magnetite mineralisation was flushed by cooler, later-stage brines that had reacted with sedimentary or metamorphic rocks, and the reduction of these brines, driven by conversion of magnetite to hematite, was suggested to have resulted in Cu–Au precipitation. A modified two-stage model with early high temperature magnetite overprinted by retrograde hematite has also been recently suggested for Prominent Hill[9].

Although not recovered economically, light REEs, in particular Ce, are enriched in IOCG deposits[7,10,11]. As early textures were largely obliterated by the massive fluid flow associated with the extensive and comprehensive hematitisation and mineralisation event in high-grade ores, we investigated the less intense partial hematitisation textures in the subeconomic deposits of the Olympic Dam IOCG province, at Torrens Dam and Emmie Bluff[25,26]. A comparison of the hematite textures in these prospects (Fig. 5) with the experimental textures for Ce-catalysed hematitisation (Fig. 1a) reveals striking similarities, with increased porosity in hematite and discontinuities existing at the hematite–magnetite boundary (Fig. 1b). This provides strong evidence that Ce was active in the hematitisation process, even in the smaller hydrothermal systems of the subeconomic deposits.

Regional faults are typically the primary control on fluid infiltration and brecciation in IOCGs[6,24]. However, hydrothermal alteration can mediate fluid circulation away from these primary infiltration points by increasing or decreasing porosity of ore and wall rocks, and has also been recognised to be the main control on breccia formation locally[10,27,28]. Positive feedback between structurally controlled fluid influx and enhanced fluid–rock interaction as a result of reaction-induced porosity results in more efficient ore deposition and prolonged duration of hydrothermal activities, both key requirements for forming large ore deposits. At Olympic Dam, the F-rich nature of the ore has been interpreted to indicate that corrosive F-rich ore fluids enhanced permeability and breccia formation[29]. However, recent experimental and theoretical data

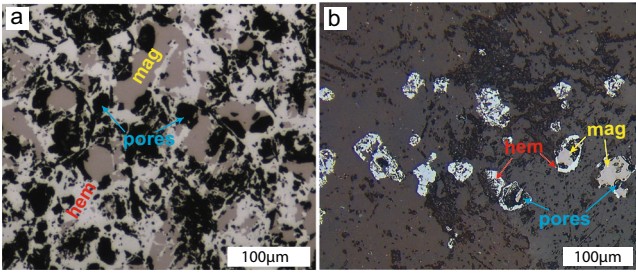

**Fig. 5 Textures of ores from the Olympic Dam IOCG province. a** Massive iron oxide ores from Torrens Dam (drill core TD 2), with hematite partially replacing magnetite. Hematite is porous with large pores. Pores are usually developed at the hematite–magnetite boundary. **b** Hematitised rock from Emmie Bluff (drill core SAE 7), with hematite partially replacing magnetite at grain rims and coarse porosity/gap developed along magnetite–hematite boundaries. Hematite grains are porous and usually display a large pore at their core, with only a few preserving some magnetite relicts at their cores.

suggest that the ore fluids carried relatively low fluoride compared to chloride and low stability of Fe–F complexes, which would limit their aggressive character in this regard[19,30].

Ce-catalysed replacement of early magnetite by hematite provides an alternative to fluorine for generating abundant porosity. This mechanism provides an explanation for the positive correlation of Cu, Au, and U with increasing hematite at Olympic Dam[11]. The Olympic Dam system is largely a chemical breccia, where the breccia texture was created by variable chemical replacement of the pre-existing rock[10,27–29,31]. Replacement of abundant pre-ore stage magnetite via Ce-catalysed hematitisation may be one of the key porosity-inducing reactions. This process also likely affects rock rheology, promoting physical brecciation and enhancing further advective fluid flow. Therefore, given the ubiquitous association between IOCG mineralisation and REE enrichment, we suggest that this dynamically enhanced syn-mineralisation porosity (Fig. 3), strongly affected by the Ce catalytic effect, is an important factor influencing the grade and scale of IOCG and REE deposits globally.

The replacement of magnetite by hematite is commonly considered to reflect the influx of oxidizing fluids[9,26]. The involvement of oxidised, sulfate-bearing fluids in the ore stage at Olympic Dam is indicated by the abundance of barite in the ores (~1.2 wt%)[11] and is consistent with the location of the orebody at or near the paleosurface[9,32]. The proposed Ce-catalysed hematitisation reaction is nominally redox-independent (reaction (2)). However, it is important to note that redox-independent hematitisation is not inconsistent with models that involve oxidised, sulfate-rich fluids. In practice, the redox-independent hematitisation process decouples hematitisation and porosity creation on the one hand, and fluid reduction and ore precipitation on the other hand. In this model, reduction of fluid-borne components occurs via reaction with aqueous Fe(II) released by the mineral replacement reaction, rather than via direct reaction with magnetite.

A number of arguments favour a predominance of redox-independent replacement of magnetite by hematite at Olympic Dam. (i) Direct oxidation of magnetite via reaction (1) involves a volume expansion of 1.66%[4], which is likely to decrease the overall porosity and permeability in orebodies, thus limiting fluid flow, fluid–mineral interaction, and ore mineral precipitation[24]. In contrast, the redox-independent reaction (2) generates 33% porosity, promoting fluid flow, brecciation, and providing space for ore mineral precipitation. (ii) The observed magnetite replacement textures and Ce enrichment strongly suggest that Ce-catalysed redox-independent replacement was widespread in the province. (iii) Sulfate is the most likely oxidant invoked for

replacement via the oxidative pathway[9]. However, a previous study also observed that even in the presence of a much stronger oxidant ($O_2$), the replacement of magnetite by hematite proceeded via a combination of redox-independent and oxidative pathways, as a result of the faster kinetics of the redox-independent pathway[4]. (iv) In addition, generation of bisulfide via sulfate reduction at the reaction front would result in rapid precipitation of Fe and base metal sulfides; yet the highly brecciated, hematite-rich core of Olympic Dam deposit is barren with respect to sulfides. Therefore, we suggest that simple oxidation reaction of magnetite to form hematite (reaction (1)) was likely not the predominant pathway for generation of high-grade, porous hematite-associated ores in IOCG deposits.

The formation of large ore deposits commonly reflects the optimal conjunction of otherwise common physical and chemical processes[33]. In this context, positive feedbacks that increase the efficiency of a given mineralisation process but require a set of highly specific conditions to become active may be important in explaining the size, distribution, and rarity of large ore deposits. In the examples of IOCG and REE deposits, our new results suggest that the pre-existence of volumetrically important magnetite alteration may have been critical in paving the way for efficient mineralisation during later Cu–Au ore stages; rapid, Ce-catalysed, redox-independent hematitisation created large amounts of porosity, thereby promoting positive feedback between fluid flow, permeability enhancement, brecciation, fluid–rock interaction, and ultimately ore deposition.

This is the first time that a trace element has been demonstrated to act as a catalyst for an ICDR reaction. Such kinetic effects are difficult to predict empirically, and their recognition requires further experimental effort, because they arise from a complex interplay between mineral solubility, nucleation, growth, and mass transfer at nano- to micro-scales. Although it is challenging to predict which elements may play a similar role in a large range of hydrothermal systems, such effects are likely to affect a number of important geological processes, including hydrous metamorphism, metasomatism, and sodic/potassic alteration. Discovery of such effects will also help with identifying new methods of ore extraction and mineral processing in the future.

## Methods

**Hydrothermal experiments.** Natural magnetite from the Itabira district, Minas Gerais, Brazil (SA Museum sample G32618), was used as the starting material. The material was crushed and grains ranging in size from 150 to 250 μm were selected for the experiment. The initial magnetite was analysed using powder X-ray diffraction (XRD) and the results show >99% $Fe_3O_4$. The solution was prepared with an acetic acid–sodium acetate buffer, to obtain a calculated pH of 4 at 25 °C. The salinity of the buffer was controlled by addition of 0.5 M NaCl. For each run, the solution was doped with 200–300 ppm La(III), Ce(III), and Nd(III), respectively. Samples were reacted at 200 °C for 8–14 days. Control experiments were conducted for two sets of solutions to quantify the effects of solution redox: (i) non-oxidative solutions, prepared by bubbling with $N_2$ gas, for which hematite and magnetite buffer the $fO_2(g)$ and (ii) oxidative solutions prepared by addition of 0.5 g 30% $H_2O_2$ solution. Details of our experiments are summarized in Supplementary Table 1.

**Textural and morphological characterisation.** Polarized optical microscopy was conducted using an Olympus BX51 microscope. SE imaging and EBSD were conducted with a FEI Quanta 3D Field Emission Scanning Electron Microscope at the Monash Centre of Electron Microscope, Monash University, Australia. Samples for SE imaging and EBSD were embedded in epoxy resin, polished, and then coated with thin carbon film (~2–4 nm). The accelerating voltage for SE was maintained at 15 kV. EBSD patterns were collected at 15 kV, 11 nA with TSL OIM EBSD system. EBSD data were analysed using TSL OIM 8 software and Matlab MTEX toolbox. Crystallographic data for hematite and magnetite were taken from the American Mineralogist Crystal Structure Database.

**Powder XRD.** Powder XRD patterns were collected on a Bruker D8 Advance Cobalt Machine (Co-Kα1 radiation, λ = 1.78892 Å) at the Monash X-ray Platform at Monash University, Australia, using 40 kV and 25 mA and a 0.6 mm slit. Phase identification was conducted using Bruker EVA software. Phase fractions (i.e.

hematite and magnetite) were determined via Rietveld refinement[34] using the Bruker Diffrac.TOPAS package. Crystal structure data of magnetite and hematite were taken from ICDD PDF-4+ database.

**Synchrotron XFM**. Elemental distribution of Ce, La, and Nd were mapped at the X-ray fluorescence spectroscopy (XFM) beamline at the Australian Synchrotron, Melbourne, Australia. The incident beam energy was set at 18.5 keV using a Si(111) monochromator with an energy resolution of $\Delta E/E$ of ~$2.8 \times 10^{-4}$. The beam was focused to a ~$2 \times 2\,\mu m^2$ spot size using Kirkpatrick-Baez mirrors. Fluorescence data were collected using the Maia model D384 detector array, which has an energy resolution of 240 eV and can detect elements down to atomic number 15 (phosphorous)[35,36]. Samples were mapped using scanning speeds ranging from 2 to 5 mm/s, corresponding to dwell times of 0.3–1.75 ms/pixel. Standard foils (Pt, Mn, Fe) were used to constrain the detector geometry and efficiency and to translate ion chamber counts to flux.

The data were analysed with the GeoPIXE software package[37,38], which utilizes the dynamic analysis method[39–41] to subtract background, unfold overlapping fluorescence peaks, and then project elemental images after fitting full fluorescence spectra (as opposed to just region-of-interest data).

The oxidation state of Ce was mapped using the XANES imaging technique, as described in Etschmann et al.[42] and Etschmann et al.[43]. XANES stacks were measured by collecting SXRF maps at 109 irregularly spaced monochromator energies that spanned the Ce-L$_3$ edge, with 0.5 eV steps across the edge. A separate dynamic analysis matrix was used for beam energy when processing the stack, in order to track the changing energy of the scatter peaks. The intensities of the Ce L$\alpha$ peak at each pixel in the SXRF map, at each monochromator energy, were extracted and used to construct XANES spectra at each pixel and integrated over regions in the map selected based on Ce L$\alpha$ intensity ratios at different energies or on sample composition.

## Data availability

The data that support the findings of this study are available from the corresponding author upon reasonable request.

## Code availability

The Matlab MTEX toolbox used for EBSD analysis is available at https://mtex-toolbox.github.io. Further support on the details of processing algorithms can be obtained from the corresponding author upon request.

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

## Acknowledgements

This study is funded by the Australian Research Council (grant DP140102765 to J.B.) and the McKinstry Fund from the Society of Economic Geologist (grant SRG_17-31 to Y.X.). The authors would like to thank Marco Fiorentini, Andy Wilde, and an anonymous reviewer for their constructive comments. The authors are grateful to Kan Li, for helping with sample collection, and Daryl Howard, for facilitating data collection at the XFM beamline at the Australian Synchrotron, part of ANSTO (proposal AS181/XFM/12890).

## Author contributions

Y.X. and J.B. designed the research. Y.X. and B.E. conducted the experiments. Y.X., J.B., B.E., and X.F. conducted the analytical work. Y.X. and J.B. wrote the manuscript with important contributions from A.G.T. and A.J.F.

## Competing interests

The authors declare no competing interests.
