## [Peer Review File · Nature Communications]

Editorial Note: This manuscript has been previously reviewed at another journal that is not operating a transparent peer review scheme. This document only contains reviewer comments and rebuttal letters for versions considered at Nature Communications .

REVIEWERS' COMMENTS

Reviewer #1 (Remarks to the Author):

Dear Editor,

I hope you are well. I have gone through the document that you sent me and I read the authors' answers to the concerns of Reviewer 2 (I was Reviewer 1 in that round).

As you will see, I was actually happy with the revised draft of the authors and I am surprised that the manuscript was ultimately rejected by Nature Geoscience.

I personally think that the authors addressed the concerns of Reviewer 2. As a result, I confirm my positive evaluation of the manuscript as well as my recommendation for acceptance, given that I think that the comments of Reviewer 2 do not unveil any fatal flaw of the study. I think that such work could benefit the scientific community in the effort to better characterise the key "ingredients" that facilitate the transport and concentration of metals in ore deposits. The idea that a trace element such as Cerium could act as a catalyst of this process is very intriguing. Explorers may think of specific settings where light rare-earth element concentrations may be enhanced, thus being able to prioritise certain areas over others.

In summary I confirm my previous review and I recommend the manuscript for acceptance in Nature Communications.

All the best

Marco Fiorentini

Reviewer #2 (Remarks to the Author):

This paper presents the results experiments on the alteration of magnetite to hematite in the presence of a redox sensitive trace element, and then compares the resulting textures to natural materials from the Olympic Dam district, Australia. The paper is well written throughout and I have no issues with the presentation of the manuscript. The role of reaction generated porosity in mineral alteration reactions is well established, but the self-regenerating catalysis role for Ce(III) to (IV) reaction alongside magnetite to hematite alteration is a novel idea, and is well supported by the data presented here. The role in ore formation may be slightly overstated as microfractures and grain boundaries will also act as fluid conduits at this scale, but nevertheless the process has potentially large influences in a range of mineral deposits types – skarns are notable example not mentioned here. The near ubiquity of magnetite and hematite in mineralising systems means that there is likely to be wide interest in the paper. There some areas where the writing could be clearer r the discussion further developed, and I highlight these points below. Otherwise this is a very thought provoking and interesting study and merits publication in Nature Communications with minor revision.

Line 27-28. Introductory paragraph. Should indicate here how the transformation of Ce(III) to Ce(IV) catalyses the replacement reaction Fe_3O_4 to Fe_2O_3 and Ce(III) are both electron loss (oxidation) reactions so hard to see how they can be coupled, expect in the presence of and additional electron acceptor (oxygen).

Line 44. Should note microfractures and grain boundaries provide an intermediate scale micro-porosity (there is not aa step in scale between macro fractures and reaction produced micro-porosity).

Line 67-69. The coupling mechanism between Ce(III/IV) and Fe(II/III) needs to be briefly reviewed here, with citations.

Line 148. Citation needed to source of equilibrium constant.

Line 157-160. The discussion is confusing here. That hematite solubility increases with decreasing pH is fine. But the next sentence then assumes constant pH, and discusses lowering of solution fO_2 by reaction 6, which does not involve free oxygen so cannot influence fO_2 . This section requires clarification.

Line 198. Say advective rather than convective here. Convection is not the only driver for fluid movement in the crust.

Line 201-204. Larger pores are argued here to facilitate fluid-mineral interaction to a greater extent than small pores. Although interconnection and permeability may be higher in large pores, surface area and porosity will be higher for small pores, so the true situation may be significantly more complex than stated.

Reviewer #3 (Remarks to the Author):

This paper reports ground-breaking research that is likely to have a significant impact in revising genetic models for Iron-Oxide Cu-Au (IOCG) and Bayan Obo type REE deposits. It is well written - I see no significant issues and recommend publishing with only very minor changes below:

L19. I suggest adding a sentence to the effect that this work shows that conversion from magnetite to hematite does not necessarily require the addition of an oxidised fluid.

L26. Probably demonstrating my ignorance, but I can't see how reaction products containing both Ce(III) and Ce(IV) indicates that Ce catalyses magnetite replacement. Suggest rephrasing.

L48. This statement requires a reference, or some further explanation.

L65. Statement says "we have noticed" yet references applies to others. Rephrase.

L158. Use of Fe(II) and Fe^{2+} . Use one or the other but not both?

L242. Discontinuity = discontinuities?

L243. Omit "result"

L272. "This reaction also affects rock rheology..." This is an assumption at this stage (and quite possibly a correct one) but I suggest toning down what is an assertion of fact.

Andy Wilde

Reply to Reviewers

Reviewer #1 (Remarks to the Author):

Dear Editor,

I hope you are well. I have gone through the document that you sent me and I read the authors' answers to the concerns of Reviewer 2 (I was Reviewer 1 in that round).

As you will see, I was actually happy with the revised draft of the authors and I am surprised that the manuscript was ultimately rejected by Nature Geoscience.

I personally think that the authors addressed the concerns of Reviewer 2. As a result, I confirm my positive evaluation of the manuscript as well as my recommendation for acceptance, given that I think that the comments of Reviewer 2 do not unveil any fatal flaw of the study. I think that such work could benefit the scientific community in the effort to better characterise the key "ingredients" that facilitate the transport and concentration of metals in ore deposits. The idea that a trace element such as Cerium could act as a catalyst of this process is very intriguing. Explorers may think of specific settings where light rare-earth element concentrations may be enhanced, thus being able to prioritise certain areas over others.

In summary I confirm my previous review and I recommend the manuscript for acceptance in Nature Communications.

All the best

Marco Fiorentini

Reply:

We are very grateful for the constructive comments from Dr. Fiorentini which have significantly improved our manuscript during the three rounds of revision.

Reviewer #2 (Remarks to the Author):

This paper presents the results of experiments on the alteration of magnetite to hematite in the presence of a redox sensitive trace element, and then compares the resulting textures to natural materials from the Olympic Dam district, Australia. The paper is well written throughout and I have no issues with the presentation of the manuscript. The role of reaction generated porosity in mineral alteration reactions is well established, but the self-regenerating catalysis role for Ce(III) to (IV) reaction alongside magnetite to hematite alteration is a novel idea, and is well supported by the data presented here. The role in ore formation may be slightly overstated as microfractures and grain boundaries will also act as fluid conduits at this scale, but nevertheless the process has potentially large influences in a range of mineral deposits types – skarns are notable example not mentioned here. The near ubiquity of magnetite and hematite in mineralising systems means that there is likely to be wide interest in the paper. There some areas where the writing could be clearer or the discussion further developed, and I highlight these points below. Otherwise this is a very thought provoking and interesting study and merits publication in Nature Communications with minor revision.

Reply:

Many thanks for the positive comments!

Line 27-28. Introductory paragraph. Should indicate here how the transformation of Ce(III) to Ce(IV) catalyses the replacement reaction Fe_3O_4 to Fe_2O_3 and Ce(III) are both electron loss (oxidation) reactions so hard to see how they can be coupled, expect in the presence of and additional electron acceptor (oxygen).

Reply:

Thanks. This is indeed puzzling at first sight, until one considers the fluid chemistry at the reaction interface, as discussed in detail in the body of the paper. We have added in the abstract: "Cerium acts as a catalyst affecting the nucleation and growth of hematite by controlling the $\text{Fe}^{2+}(\text{aq})/\text{Fe}^{3+}(\text{aq})$ ratio at the reaction interface". (NB. abstract now follows the 150 words limit of Nat.Com.).

Line 44. Should note microfractures and grain boundaries provide an intermediate scale micro-porosity (there is not a step in scale between macro fractures and reaction produced micro-porosity).

Reply:

Thanks. We have followed the reviewer's suggestion and modified the text to read "away from these, *micro-fractures and grain boundaries were thought to be the main fluid pathways until it was realised that porosity is intrinsically generated during many fluid-mediated mineral replacement reactions*". This highlights the fairly recent change in paradigm around porosity as a result of the understanding of the nature of reaction mechanism over the past 15 years.

Line 67-69. The coupling mechanism between Ce(III/IV) and Fe(II/III) needs to be briefly reviewed here, with citations.

Reply:

This has been rephrased: "*for example, preferential Ce removal from seawater results from adsorption of dissolved Ce(III) onto Fe- (and Mn-) oxides followed by oxidation of the surface complex*"¹¹.

Line 148. Citation needed to source of equilibrium constant.

Reply:

This has been updated: "*(based on the SUPCRT database*¹⁶*, updated with the properties of CeO₂(s) of Konings, Beneš*¹⁷*".*

Line 157-160. The discussion is confusing here. That hematite solubility increase with decreasing pH is fine. But the next sentence then assumes constant pH, and discusses lowering of solution fO₂ by reaction 6, which does not involve free oxygen so cannot influence fO₂. This section requires clarification.

Reply:

We have improved the wording: "*Conversely, reactions [4] and [5] show that hematite precipitation is hindered at low pH and/or under reducing conditions (i.e., high Fe²⁺/Fe³⁺) in solution*"¹⁶. *These conditions are favoured by local Ce³⁺ oxidation following reaction [6]*"

Line 198. Say advective rather than convective here. Convection is not the only driver for fluid movement in the crust.

Reply:

Thanks. This has been updated.

Line 201-204. Larger pores are argued here to facilitate fluid-mineral interaction to a greater extent than small pores. Although interconnection and permeability may be higher in large pores, surface area and porosity will be higher for small pores, so the true situation may be significantly more complex than stated.

Reply:

Thanks. The porosity textures clearly indicate different fluid transport mechanisms; we have added "*as illustrated by the greater reaction extent in the presence of Ce compared to Ce-free experiments (runs MH39/MH44 vs. MH38,40,45; Supplementary Table 1)*" to address the reviewer's comment, as reaction extents (kinetics) are a direct result of this change.

Reviewer #3 (Remarks to the Author):

This paper reports ground-breaking research that is likely to have a significant impact in revising genetic models for Iron-Oxide Cu-Au (IOCG) and Bayan Obo type REE deposits. It is well written - I see no significant issues and recommend publishing with only very minor changes below:

L19. I suggest adding a sentence to the effect that this work shows that conversion from magnetite to hematite does not necessarily require the addition of an oxidised fluid.

Reply:

Many thanks for the positive comments! We have updated the abstract by mentioning the "*redox-independent (no external oxidant)*" nature of the magnetite to hematite transformation.

L26. Probably demonstrating my ignorance, but I can't see how reaction products containing both Ce(III) and Ce(IV) indicates that Ce catalyses magnetite replacement. Suggest rephrasing.

Reply:

We have simplified the abstract to fit the 150 words limit – this statement has been removed.

L48. This statement requires a reference, or some further explanation.

Reply:

Added two references.

L65. Statement says “we have noticed” yet references applies to others. Rephrase.

Reply:

This has been updated: “*it is commonly observed.*”

L158. Use of Fe(II) and Fe²⁺. Use one or the other but not both?

Reply:

Good catch on the inconsistent formatting. We now use Roman numerals for oxidation states, and 2+/3+ for the aqueous ions.

L242. Discontinuity = discontinuities?

Reply:

This has been updated.

L243. Omit “result”

Reply:

This has been corrected.

L272. “This reaction also affects rock rheology... “ This is an assumption at this stage (and quite possibly a correct one) but I suggest toning down what is an assertion of fact.

Reply:

This has been updated: “*This process also likely affects rock rheology*”.